# Prognostic Impact of Serum Albumin for Developing Heart Failure Remotely after Acute Myocardial Infarction

**DOI:** 10.3390/nu12092637

**Published:** 2020-08-29

**Authors:** Goro Yoshioka, Atsushi Tanaka, Kensaku Nishihira, Yoshisato Shibata, Koichi Node

**Affiliations:** 1Department of Cardiovascular Medicine, Saga University, Saga 849-8501, Japan; tanakaa2@cc.saga-u.ac.jp (A.T.); node@cc.saga-u.ac.jp (K.N.); 2Miyazaki Medical Association Hospital Cardiovascular Center, Miyazaki 880-0834, Japan; nishihira@med.miyazaki-u.ac.jp (K.N.); yshibata@cure.or.jp (Y.S.)

**Keywords:** nutritional status, acute myocardial infarction, albumin, heart failure

## Abstract

Low serum albumin (LSA) on admission for acute myocardial infarction (AMI) is related to adverse in-hospital outcomes. However, the relationship between LSA and long-term post-AMI cardiovascular outcomes is unknown. A single-center, non-randomized, retrospective study was performed to investigate the prognostic impact of LSA at admission for AMI on cardiovascular death or newly developed HF in the remote phase after AMI. Admission serum albumin tertiles (<3.8, 3.8–4.2, ≥4.2 g/dL) were used to divide 2253 consecutive AMI from February 2008 to January 2016 patients into three groups. Primary outcome was a composite of hospitalization for HF and cardiovascular death remotely after AMI. Cox proportional hazard models were used to explore the relationship between admission LSA and primary outcome. During follow-up (median: 3.2 years), primary composite outcome occurred in 305 patients (13.5%). Primary composite outcome occurred individually for hospitalization for HF in 146 patients (6.5%) and cardiovascular death in 192 patients (8.5%). The cumulative incidence of primary composite outcome was higher in the LSA group than the other groups (log-rank test, *p* < 0.001). Even after adjustments for relevant clinical variables, LSA (<3.8 mg/dL) was an independent predictor of remote-phase primary composite outcome, irrespective of the clinical severity and subtype of AMI. Thus, LSA on admission for AMI was an independent predictor of newly developed HF or cardiovascular death and has a useful prognostic impact even remotely after AMI.

## 1. Introduction

In the evolution of treatment for acute myocardial infarction (AMI), early reperfusion therapies have reduced short-term mortality [1]. Especially in the era of primary percutaneous coronary intervention (PCI), a lower in-hospital mortality rate has become well established in patients with AMI. However, heart failure (HF) as a remote phase complication after AMI is still an important issue because it worsens the chronic prognosis. Hence, risk stratification for its prevention should be carried out in the early phase [2,3,4,5]. Previous reports have suggested a number of possible predictors of cardiovascular events after AMI, including patient background, electrocardiographic features, factors of reperfusion therapy including onset-to-balloon time, ventricular dysfunction, frailty, and poor nutritional status [6,7,8]. Serum albumin is one of the most important nutritional indicators. Among several others, including prognostic nutritional index (PNI), geriatric nutritional risk index (GNRI), and controlling nutritional status score (CONUT), serum albumin is a simple indicator which has been widely used as a quantitative measure of nutritional status or inflammatory reaction. Low serum albumin (LSA) has been reported to be associated with a number of cardiovascular diseases and disabilities [9,10]. In patients with AMI, the presence of LSA has been associated with in-hospital mortality and development of HF during hospitalization [8]. However, whether LSA at admission is a risk factor for the development of HF in the remote phase after AMI is unknown, and the precise impact on long-term mortality also remains uncertain. We therefore investigated the prognostic impact of LSA at admission for AMI on cardiovascular death or newly developed HF in the remote phase after AMI.

## 2. Materials and Methods 

### 2.1. Patient Population

The present study was a single-center, non-randomized, retrospective study performed in the Miyazaki Medical Association Hospital, Japan. A total of 2266 consecutive AMI patients with either ST elevated or non-ST elevated myocardial infarction (STEMI and NSTEMI, respectively) were recruited from February 2008 to January 2016 (Figure 1). Thirteen patients were excluded from this analysis for missing data on serum albumin at admission. All patients provided informed consent for both the procedure and the subsequent data collection and analysis for research purposes. Ethics approval was obtained from the Institution Review Board of Miyazaki Medical Association Hospital (2019-23).

### 2.2. Definition and Diagnosis of STEMI and NSTEMI

Clinical diagnoses of STEMI and NSTEMI, based on the 2007 universal definitions [9], were made by the treating cardiologists. In brief, STEMI and NSTEMI were redefined as follows: for STEMI, patients had to have chest symptoms, ST-segment elevation in two contiguous leads or left bundle branch block, and an elevated biochemical marker of myocardial necrosis (high-sensitivity troponin T >0.032 ng/mL or creatine phosphokinase (CPK) at least two-fold the upper limit of normal), whereas for NSTEMI, patients had to have chest symptoms, ST-segment depression or T-wave inversion in two contiguous leads, and an elevated biochemical marker of myocardial necrosis. Unstable angina pectoris patients that did not fulfill either definition in the present study were excluded. The therapeutic strategies for AMI treatment depended on the practice of each individual cardiologist, but all patients’ treatments followed the guidelines set forth by the Japanese Circulation Society and American College of Cardiology (ACC)/ American Heart Association (AHA) for the diagnosis and treatment of AMI [3]. 

### 2.3. Data Collection and Outcome

The following types of data were collected: baseline demographics and clinical characteristics of study patients, medical history, presenting signs and symptoms, results of blood tests, transthoracic echocardiography, electrocardiography, cardiac procedures, and clinical outcome. Transthoracic echocardiography was performed for all patients immediately after admission and left ventricular ejection fraction (LVEF) was estimated by the standard biplane Simpson method. In addition, all blood biomarkers were measured within 24 h after admission. Clinical follow-up was achieved through clinic visits, telephone calls, and records from hospital admissions. 

The primary outcome was a composite of hospitalization for HF or cardiovascular death. The diagnosis of HF was made based on the guidelines, in which HF is diagnosed by the presence of at least one sign (rales, peripheral edema, ascites, or radiographic evidence of pulmonary congestion) and one symptom (dyspnea, orthopnea or edema), regardless of ejection fraction [11]. The secondary outcomes included the individual components of primary composite outcome, all-cause death, and in-hospital death.

### 2.4. Statistics

For continuous variables, normally distributed data are reported as the mean ± standard deviation; nonparametric data are reported as the median and interquartile range (IQR). For categorical variables, data are presented as counts and percentages. Comparisons of continuous variables between groups were performed by Student’s t-test or Mann–Whitney U test, as appropriate. Comparisons of categorical variables were assessed by the chi-squared or Fisher exact test, as appropriate. The cumulative incidence of each outcome was calculated according to the Kaplan–Meier method. The effects of serum albumin on primary outcome were determined using multivariate Cox proportional hazard regression analysis. Patients were stratified to tertiles based on serum albumin levels (<3.8, 3.8–4.2, and ≥4.2 g/dL) at admission. Especially, the receiver operating characteristic curve of albumin for primary composite outcome showed that lower cut-off value was also 3.8 g/dL. Univariate and multivariate analyses using the Cox model were performed to determine the relationships between the albumin level and clinical outcome, independent of the following confounders: Model 1 (age, sex), Model 2 (Model 1 plus body mass index (BMI) and coronary risk factors, Model 3 (Model 2 plus onset-to-admission time, pre-thrombolysis in myocardial infarction (TIMI) grade, percutaneous coronary intervention (PCI), max creatine kinase, length of hospital stay and statin use at discharge) and Model 4 (Model 3 plus other medication use at discharge (antiplatelet, β-blocker, angiotensin-converting enzyme inhibitor, angiotensin II receptor blocker, mineralocorticoid receptor antagonist and diuretic), left ventricular ejection fraction at acute phase, cardiogenic shock, high-sensitivity troponin T level, C reactive protein, alanine aminotransferase as liver function and Killip ≥III. In order to further clarify the impact of serum albumin, we also classified patients according to the presence or absence of the following risk factors for primary outcome, which were estimated by multivariate analysis and receiver operating characteristic curve analysis: brain natriuretic peptide (BNP) >200 pg/mL, peak CPK >8000 IU/L, estimated glomerular filtration rate (eGFR) <30 mL/min/1.73 m^2^, and LVEF <35% in the acute phase. Furthermore, Cox proportional hazards analysis for primary composite outcome in each subtype of AMI (STEMI or NSTEMI) was also performed. A two-sided *p* value <0.05 was considered statistically significant. All statistical analysis was performed using JMP^®^ 14 (SAS Institute Inc., Cary, NC, USA).

## 3. Results

### 3.1. Patient Characteristics on Admission

The baseline demographics and clinical characteristics of patients are shown in Table 1. The mean patient age was 70.1 ± 12.7 years old, with 69.7% being male. Electrocardiography revealed that 69.2% were STEMI and 30.8% were NSTEMI. Patients in the lowest albumin tertile (<3.8 g/dL) were older and more likely to have lower body mass index, lower systolic blood pressure, and more frequent histories of diabetes mellitus, old myocardial infarction, and malignancy. This patient group also had lower levels of hemoglobin, eGFR, and lipid parameters and higher levels of high-sensitivity troponin T and BNP, compared to the other two groups.

### 3.2. Procedures and Management after AMI

Detailed information on AMI procedures and clinical management during hospitalization is shown in Table 1. The onset-to-admission time in the lowest albumin group was significantly longer than in the other groups and delayed post-AMI arrival (≥48 h from onset) was more frequent in this group. An incident Killip class ≥III was also observed more frequently in this group, with LVEF being lower than in the other groups. Overall, 90.9% of patients underwent revascularization (87.2% PCI and 3.6% coronary artery bypass graft (CABG)), with the percentage of patients who underwent revascularization being significantly lower in the lowest albumin group. On the other hand, patients in the lowest albumin group needed longer hospital stays and more frequent mechanical supports, such as intra-aortic balloon pumping and extracorporeal membrane oxygenation.

Regarding medications at discharge, the prevalence of antiplatelet, statin, and angiotensin-converting enzyme inhibitor (ACE-I) prescriptions was lower in the lowest albumin group, while mineralocorticoid receptor antagonist and diuretic prescriptions were more prevalent in this group.

### 3.3. Clinical Outcome

The median duration of follow-up was 3.2 (IQR, 1.6–5.4) years. The primary composite outcome of hospitalization for HF or cardiovascular death occurred in 305 patients (13.5%); individual components of the primary composite outcome occurred in 146 patients (6.5%) for hospitalization for HF and 192 patients (8.5%) for cardiovascular death (Table 1). In addition, the secondary outcomes of all-cause death and in-hospital death were observed for 375 patients (16.6%) and 154 patients (6.8%), respectively (Table 1). Kaplan-Meier curves of clinical outcomes in patients stratified by tertile of serum albumin level on admission are presented in Figure 2. Cumulative incidences of primary composite outcome and secondary outcomes were significantly higher in the lowest albumin group than in the other two groups (log-rank test, *p* < 0.001 for each outcome). In the Cox proportional hazards model analyses (Table 2), the incidences of primary composite outcome and other outcomes were significantly and incrementally higher in the lowest and middle albumin groups than in the highest group (albumin ≥4.2 g/dL: reference). Even after adjusting for relevant clinical variables including LVEF and cardiogenic shock, all medication uses at discharge, high-sensitivity troponin T level, C reactive protein, alanine aminotransferase as liver function and Killip ≥III, these associations between cardiac events and lower tertile of serum albumin remained significant (Table 2). 

### 3.4. Subgroup Analysis for Primary Composite Outcome

Receiver operating characteristic curve analyses of clinical variables (BNP, peak CPK, eGFR, LVEF, albumin and high-sensitivity troponin T) for primary composite outcome were performed (Figure 3) and determined corresponding cut-off values. Depending on the presence or absence of the estimated risk factors for primary outcome (AMI due to left main trunk, BNP >200 pg/mL, peak CPK >8000 IU/L, eGFR <30 mL/min/1.73 m^2^, and LVEF <35%), 611 patients who had at least one of these risk factors on admission were identified as a high-risk group, and 1642 patients without these risk factors on admission were identified as a non-high-risk group. The incidence of primary composite outcome occurred in 145 patients (23.7%) in the high-risk group and 160 patients (9.7%) in the non-high-risk group. In the high-risk group, the incidence of primary composite outcome was significantly higher in the lowest albumin group than in the highest group (albumin ≥4.2 g/dL: reference) even after multiple adjustments by relevant clinical factors (Table 3). Finally, even in the non-high-risk group, the lowest albumin group exhibited significantly higher rates of primary composite outcome compared with the corresponding reference subgroups (albumin ≥4.2 g/dL) in the analyses for the crude model, Model 1, Model 2, and Model 4 (Table 3).

Regarding subtype of AMI, the incidence of primary composite outcome occurred in 229 patients (14.7%) of 1558 STEMI patients and 76 patients (10.9%) of 695 NSTEMI patients. In each subtype of AMI, the incidence of primary composite outcome was significantly higher in the lowest albumin group than in the reference highest group even after multiple adjustments by relevant clinical factors (Table 3). 

## 4. Discussion

Our present study clearly demonstrated that LSA (<3.8 g/dL) on hospital admission was an independent predictor of newly developed HF or cardiovascular death in the remote phase after AMI, both in the overall study cohort and even in the non-high-risk patients who did not have any of the risk factors associated with clinical severity of AMI. Furthermore, these clinical impacts were also demonstrated in the STEMI and the NSTEMI subgroups. In addition, LSA adversely affected both short-term and long-term mortalities. These findings suggest that the LSA on admission for AMI may have a useful prognostic impact in the remote phase after AMI; even after discharge, especially careful follow-up is therefore needed for AMI patients with LSA.

Previous studies reported that risk factors for increased short-term mortality in AMI patients included advanced age, prior MI, female sex, Killip class III or IV, left anterior descending coronary artery (LAD) involvement, complete occlusion of the infarct vessel at baseline, severely reduced LVEF, frailty, and nutritional status in the acute phase [12,13,14]. Several nutritional indicators including PNI, GNRI, and CONUT score have been reported as useful predictors of prognosis in patients with cardiovascular disease [15,16,17]. PNI is independently associated with long-term survival in patients hospitalized for acute heart failure with either reduced or preserved LVEF [15]. GNRI is a significant prognostic factor in clinical outcomes after AMI during hospitalization [16]. In addition, CONUT score demonstrated prognostic impact of nutritional status in STEMI patients [17]. PNI is calculated using serum albumin and total lymphocyte count; GNRI is calculated using serum albumin and BMI; and CONUT score is calculated using the serum albumin level, total cholesterol level, and total lymphocyte counts. All these indicators take albumin into account but require multiple values and complicated calculations. We therefore investigated the prognostic impact of serum albumin alone, a simple nutritional indicator, for cardiovascular events in the remote phase after AMI.

Serum albumin, the most abundant protein in plasma, is the main determinant of plasma oncotic pressure and the main modulator of fluid distribution between body compartments [18]. In clinical practice, it is recognized as a simple and important nutritional indicator and widely used to monitor the clinical course of several diseases and treatments, including hypovolemia, shock, burns, surgical blood loss, trauma, hemorrhage, cardiopulmonary bypass, acute respiratory distress syndrome, hemodialysis, nutritional support, and resuscitation [18,19,20]. 

In the context of cardiovascular disease, several studies have reported that LSA is associated with prognosis. In heart failure patients, LSA has proven to be a useful predictor. Regardless of the etiology of heart failure, chronic heart failure patients with LSA were revealed to have increased long-term mortality [21,22]. LSA was also an independent predictor of long-term event in patients with acute heart failure [23]. In these reports, the negative impact of LSA was thought to result from malnutrition, decreased hepatic synthesis, increased vascular permeability, and/or renal failure. 

For stable coronary artery disease patients, LSA predicted adverse events including all-cause death, stroke, and myocardial infarction during long-term follow-up [24,25]. In these reports, the main cause of the negative effect was thought to be severe atherogenesis-related inflammation. Recently, the relationship between LSA and prognosis after AMI was reported [26,27], suggesting that LSA is associated with new onset HF during AMI hospitalization and resultant poor prognosis after AMI. For these reports, new onset HF during AMI hospitalization seemed to be main factor of adverse outcome.

LSA may facilitate increased peripheral edema and pulmonary congestion even at lower left atrial pressures, acting as an aggravating factor of heart failure [28]. It has recently been reported that inflammation has a negative impact on serum albumin level, making it an important inflammatory marker [8]. Experimental studies showed that activation of cardiac inflammation provokes LV remodeling and LV dysfunction [29]. Pathophysiological analysis also shows that excess inflammation contributes to LV remodeling associated with newly developing HF [30]. Inflammation is accelerated in the acute phase of AMI, which causes a temporary decreasing effect on the albumin level [18], then inflammation and LSA can additively and adversely affect the outcome, including HF. Thus, it is reasonable that LSA in the acute phase has a negative impact for HF and mortality in the acute phase [8]. However, the relationship between LSA at AMI admission and long-term outcomes after AMI has been unknown. If LSA at admission has an impact for prognosis of AMI developing in the remote phase, LSA can become a potent therapeutic target for prevention of HF and improvement of prognosis after AMI. 

Even after multiple adjustments by relevant clinical factors including inflammatory biomarker and hemodynamic status, our present study demonstrates the prognostic impact of LSA in AMI patients in the remote phase after AMI. Furthermore, LSA was adversely associated with long-term outcomes even in those patients who did not have any risk factors associated with clinical severity of AMI. Note that our results regarding AMI patients with LSA should help to identify at-risk patients and should promote future research regarding the effects of serum albumin levels. Of course, the questions remain whether LSA mechanistically facilitates newly developing heart failure and excess mortality or is merely a disease biomarker. 

Some limitations must be taken into account. First, this was a non-randomized, retrospective, observational study carried out in a single center between February 2008 and January 2016. Therefore, the extent to which the data apply only to the immediate environment of the patients is unknown. In addition, although revascularization therapy (including PCI and CABG) and oral medication delivery was performed based on the local treatment guidelines, nation-wide or international data could be of great value in assessing the generalizability of our findings. Second, decision-making regarding hospitalization for HF was the choice of the treating physician. It was based on at least one sign or symptom of worsening HF but did not require the use of intravenous diuretics or increase of oral diuretic dose. Therefore, this outcome might not be assessed in a completely objective manner, and there might be some uncertainty with respect to its ascertainment. Third, the optimal pharmacological therapies after AMI were also not completely achieved in the present study cohort with statins for 83.5%, ACE-I or angiotensin II receptor blockers for 66.0%, and β-blockers for 47.0%, which might affect the outcomes in the remote phase after AMI. In particular, a relatively small percentage of the present study cohort was treated with β-blockers, due mainly to low blood pressure and low heart rate. Additionally, we did not collect detailed data on the drugs administered or changed after discharge. Fourth, 69.2% of the AMI were STEMIs in the present study, which could obscure the possibility of different impacts for outcome due to different types of AMI. Fifth, serum albumin could be affected by systemic conditions resulting from the acute phase of AMI. Further research would be required to further understand the effect of changes in serum albumin levels on outcomes in the chronic phase after AMI.

## 5. Conclusions

LSA on admission for AMI was an independent predictor of adverse clinical events, including hospitalization for HF and cardiovascular death, in the remote phase after AMI, irrespective of the clinical severity and subtype of AMI.

## Figures and Tables

**Figure 1 nutrients-12-02637-f001:**
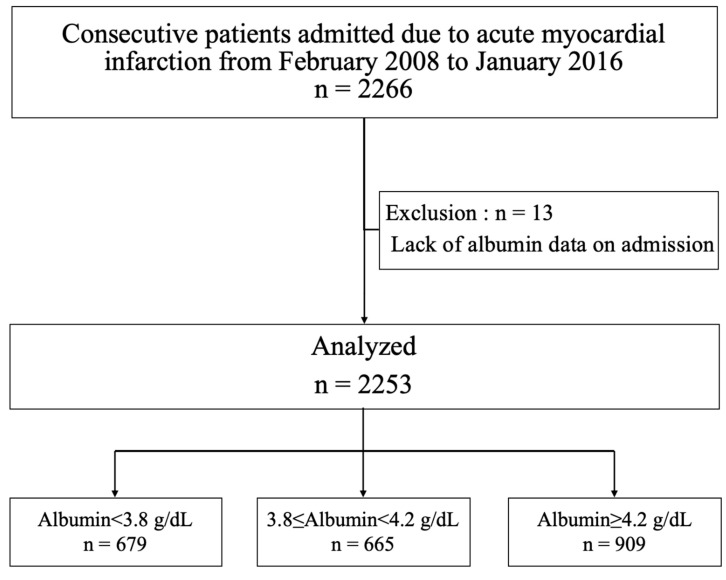
Flow diagram of the study cohort.

**Figure 2 nutrients-12-02637-f002:**
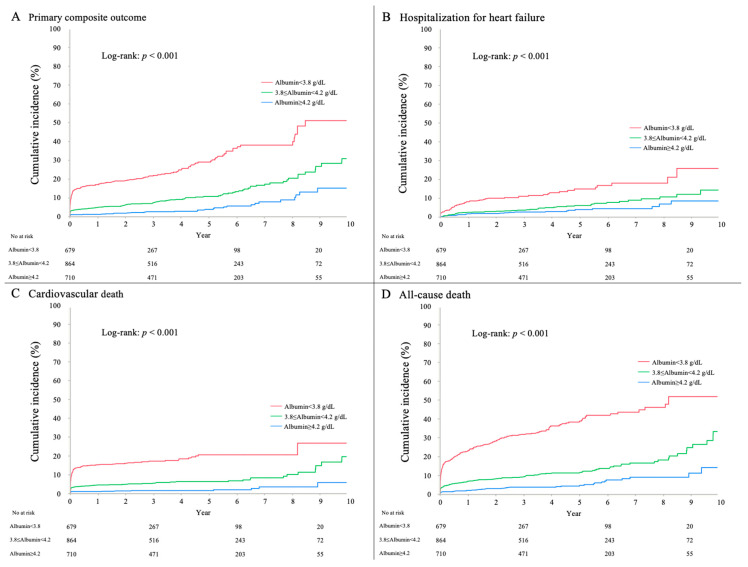
Primary and secondary outcomes. Kaplan-Meier curves show the incidence of primary composite outcome (**A**), hospitalization for heart failure (**B**), cardiovascular death (**C**), and all-cause death (**D**).

**Figure 3 nutrients-12-02637-f003:**
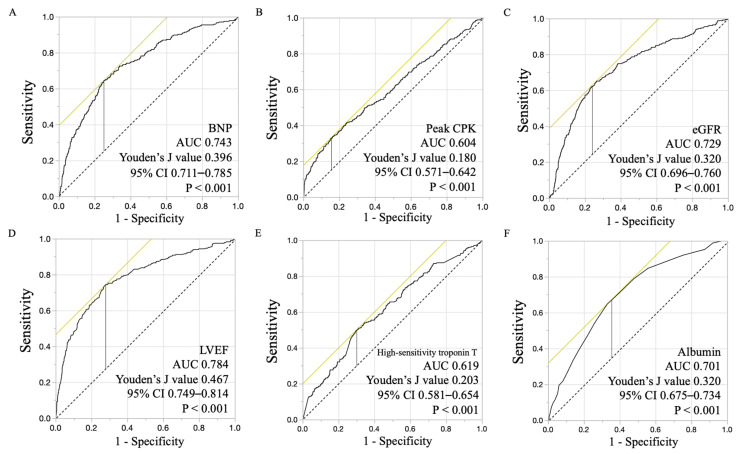
Receiver operating characteristic curve with area under the curve (AUC) of primary composite outcome: brain natriuretic peptide (BNP) (**A**), peak creatine phosphokinase (CPK) (**B**), estimated glomerular filtration rate (eGFR) (**C**), left ventricular ejection fraction (LVEF) (**D**), high-sensitivity troponin T (**E**) and albumin (**F**).

**Table 1 nutrients-12-02637-t001:** Baseline demographics and characteristics of patients.

Variables	Total	Alb < 3.8 g/dL	3.8 ≤ Alb < 4.2 g/dL	Alb ≥ 4.2 g/dL	*p*
	(*n* = 2253)	(*n* = 679)	(*n* = 665)	(*n* = 909)	
Male	1571 (69.7)	397 (58.5)	470 (70.7)	704 (77.5)	<0.001
Age, years	70.1 ± 12.7	77.0 ± 10.9	71.4 ± 11.5	64.1 ± 12.0	<0.001
Body mass index, kg/m^2^	23.7 ± 3.8	22.4 ± 3.6	23.7 ± 3.7	24.7 ± 3.7	<0.001
Heart rate, /min	79.0 ± 21.4	80.9 ± 26.0	77.4 ± 21.2	78.7 ± 17.3	0.019
Systolic blood pressure, mm Hg	137.9 ± 31.2	127 ± 33	136 ± 30	147 ± 28	<0.001
Medical history					
Hypertension	1568 (69.6)	494 (72.8)	446 (67.1)	628 (69.1)	0.069
Dyslipidemia	1068 (47.4)	243 (35.8)	314 (47.2)	511 (56.2)	<0.001
Diabetes mellitus	738 (32.8)	250 (36.8)	205 (30.8)	283 (31.1)	0.027
Smoking	981 (43.5)	230 (33.9)	293 (44.0)	458 (50.4)	<0.001
Family history of cardiovascular disease	194 (8.6)	38 (5.6)	68 (10.2)	88 (9.7)	<0.001
Myocardial infarction	115 (5.1)	53 (7.8)	36 (5.4)	26 (2.9)	<0.001
Malignancy	87 (3.9)	37 (5.5)	25 (3.8)	25 (2.8)	0.023
WBC, ×10^3^/mL	92 (56.4–120)	96 (72–126)	91 (70–117)	90 (69–118)	0.315
Hemoglobin, g/dL	13.3 ± 2.3	11.8 ± 2.2	13.3 ± 1.9	14.5 ± 1.8	<0.001
eGFR, mL/min/1.73 m^2^	63.1 ± 23.9	51.8 ± 23.9	62.0 ± 22.9	72.3 ± 20.5	<0.001
Alanine aminotransferase, U/L	22.0 (15–35)	24.0 (17–37)	21.0 (15–31)	22.5 (17–37)	<0.001
Triglyceride, mg/dL	107 (74–158)	87 (65–117)	103 (74–148)	132 (89–200)	<0.001
Total-cholesterol, mg/dL	194.5 ± 48.9	172.6 ± 43.8	194.2 ± 42.6	210.3 ± 50.7	<0.001
LDL-cholesterol, mg/dL	121.3 ± 37.9	105.0 ± 33.4	121.8 ± 35.7	132.5 ± 38.4	<0.001
HDL-cholesterol, mg/dL	46.6 ± 13.4	43.9 ± 13.7	67.4 ± 13.3	48.6 ± 13.3	0.248
High-sensitivity troponin T, ng/mL (upper limit of normal: 0.032)	0.33 (0.05–2.07)	1.25 (0.14–8.47)	0.27 (0.06–2.00)	0.15 (0.03–1.15)	<0.001
Brain natriuretic peptide, pg/mL (upper limit of normal: 18.4)	77.8 (24.0–287.2)	335.9 (97.7–790.5)	73.1 (26.9–221.8)	34.9 (14.5–93.9)	<0.001
STEMI	1558 (69.2)	624 (68.7)	471 (70.8)	463 (68.2)	0.526
NSTEMI	695 (30.8)	285 (31.3)	194 (29.2)	216 (31.8)	
Onset-to-admission time, min	275 (160–632)	335 (184–1071)	260 (160–600)	246 (159–503)	<0.001
Delayed arrival (≥48 h from onset)	64 (2.8)	29 (4.3)	20 (3.0)	15 (1.7)	<0.001
Killip class ≥3	250 (11.5)	166 (25.5)	64 (9.9)	20 (2.3)	<0.001
LVEF, %	55.3 ± 13.1	50.3 ± 14.6	56.1 ± 12.5	58.3 ± 11.1	<0.001
Pre-TIMI grade 0.1	1181 (52.4)	335 (49.3)	343 (51.6)	499 (54.9)	0.346
Peak creatine kinase, IU/L	1423 (499–3250)	1077 (372–2588)	1522 (593–3460)	1620 (517–3382)	0.118
Revascularization	2048 (90.9)	574 (84.0)	623 (93.7)	851 (93.6)	<0.001
PCI	1965 (87.2)	540 (79.5)	597 (89.8)	828 (91.1)	<0.001
CABG	83 (3.6)	34 (5.0)	26 (4.0)	23 (2.7)	<0.001
IABP	286 (12.7)	151 (22.8)	68 (10.4)	67 (7.3)	<0.001
ECMO	64 (2.8)	40 (5.9)	11 (1.7)	13 (1.4)	<0.001
Length of hospital stay, days	15 (12–20)	17 (11–26)	15 (12–21)	14 (12–17)	<0.001
Medication at discharge					
Antiplatelet	2026 (96.5)	537 (93.4)	616 (97.5)	873 (97.5)	<0.001
Statin	1753 (83.5)	435 (75.9)	511 (80.9)	807 (90.2)	<0.001
β-blocker	987 (47.0)	272 (47.5)	281 (44.5)	434 (48.5)	0.288
ACE-I	534 (25.4)	107 (18.7)	162 (25.6)	265 (29.6)	<0.001
ARB	853 (40.6)	213 (37.2)	277 (43.8)	363 (40.6)	0.063
MRA	258 (12.3)	125 (21.8)	83 (13.1)	50 (5.6)	<0.001
Diuretic	475 (22.6)	227 (39.6)	145 (22.9)	103 (11.5)	<0.001
Primary composite outcome	305 (13.5)	168 (24.7)	91 (13.7)	46 (5.1)	<0.001
Hospitalization for heart failure	146 (6.5)	70 (10.3)	46 (6.9)	30 (3.3)	<0.001
Cardiovascular death	192 (8.5)	116 (17.1)	53 (8.0)	23 (2.5)	<0.001
Secondary outcome, n (%)					
All-cause death	375 (16.6)	227 (33.4)	97 (14.6)	51 (5.6)	<0.001
In-hospital death	154 (6.8)	107 (15.8)	33 (5.0)	14 (1.5)	<0.001

Data for categorical variables given as numbers (%); data for continuous variables given as means ± standard deviation for normal distribution or medians (interquartile range) for skewed distribution. STEMI, ST elevation myocardial infarction; NSTEMI, non-ST elevation myocardial infarction; MVD, multi-vessel disease; TIMI, thrombolysis in myocardial infarction; WBC, white blood cell; eGFR, estimated glomerular filtration rate; LDL, low-density lipoprotein; HDL, high-density lipoprotein; LVEF, left ventricular ejection fraction; PCI, percutaneous coronary intervention; CABG, coronary artery bypass grafting; IABP, intra-aortic balloon pumping; ECMO, extracorporeal membrane oxygenation; ACE-I, angiotensin-converting enzyme inhibitor; ARB, angiotensin II receptor blocker; MRA, mineralocorticoid receptor antagonist.

**Table 2 nutrients-12-02637-t002:** Cox proportional hazards analysis for primary and secondary outcomes.

Outcomes	Albumin < 3.8 g/dL			3.8 ≤ Albumin < 4.2 g/dL			Albumin ≥ 4.2 g/dL (Reference)
	HR	95% CI	*p*	HR	95% CI	*p*	HR
Primary composite outcome							
Crude	4.89	1.64–6.67	<0.001	2.70	1.92–3.80	<0.001	1.00
Model 1	4.69	3.29–6.81	<0.001	1.85	1.29–2.70	<0.001	1.00
Model 2	4.52	3.11–6.63	<0.001	1.65	1.12–2.47	0.010	1.00
Model 3	3.44	1.97–6.90	<0.001	1.81	1.05–3.20	0.033	1.00
Model 4	2.94	1.37–6.51	<0.005	2.84	1.41–5.93	0.003	1.00
Hospitalization for heart failure							
Crude	3.12	2.06–4.74	<0.001	2.10	1.33–3.28	<0.001	1.00
Model 1	3.12	1.93–5.15	<0.001	1.56	0.96–2.55	0.068	1.00
Model 2	3.12	1.89–5.25	<0.001	1.56	0.94–2.60	0.079	1.00
Model 3	4.03	2.04–8.19	<0.001	1.97	1.03–3.87	0.040	1.00
Model 4	3.69	1.56–9.15	0.003	2.55	1.15–5.92	0.021	1.00
Cardiovascular death							
Crude	6.75	4.37–10.4	<0.001	3.15	1.95–5.09	<0.001	1.00
Model 1	6.15	3.84–10.2	<0.001	2.12	1.29–3.59	0.003	1.00
Model 2	5.80	3.54–9.85	<0.001	1.71	1.01–2.99	0.048	1.00
Model 3	5.07	2.01–13.8	<0.001	2.00	0.79–5.06	0.142	1.00
Model 4	4.77	1.39–18.4	0.012	2.14	0.64–7.11	0.211	1.00
All-cause death							
Crude	5.96	4.47–7.94	<0.001	2.60	1.88–3.59	<0.001	1.00
Model 1	5.61	4.06–7.88	<0.001	1.80	1.27–2.58	<0.001	1.00
Model 2	5.55	3.93–7.99	<0.001	1.69	1.16–2.48	0.006	1.00
Model 3	5.30	3.26–8.86	<0.001	1.79	1.08–3.03	0.023	1.00
Model 4	3.99	2.19–7.45	<0.001	1.70	1.01–2.90	0.045	1.00
In-hospital death							
Crude	10.2	5.91–17.7	<0.001	3.22	1.74–5.97	<0.001	1.00
Model 1	8.16	4.64–15.4	<0.001	2.21	1.18–4.37	0.013	1.00
Model 2	1.41	1.25–1.59	<0.001	0.97	0.87–1.82	0.603	1.00
Model 3	1.26	1.09–1.46	0.002	0.96	0.85–1.08	0.485	1.00
Model 4	3.00	1.11–8.66	<0.001	0.80	0.24–0.91	0.541	1.00

CI, confidence interval; HR, hazard risk. Model 1; adjusted for age and sex. Model 2; adjusted for Model 1 plus body mass index and coronary risk factors (hypertension, dyslipidemia, diabetes mellitus, smoking, family history of cardiovascular disease). Model 3; adjusted for Model 2 plus onset-to-admission time, pre-TIMI grade, percutaneous coronary intervention, max creatine kinase, length of hospital stay and statin use at discharge. Model 4; adjusted for Model 3 plus other medication use at discharge (antiplatelet, β-blocker, angiotensin-converting enzyme inhibitor, angiotensin II receptor blocker, mineralocorticoid receptor antagonist and diuretic), left ventricular ejection fraction at acute phase, cardiogenic shock, high-sensitivity troponin T level, C reactive protein, alanine aminotransferase as liver function and Killip ≥III.

**Table 3 nutrients-12-02637-t003:** Cox proportional hazards analysis for primary composite outcome in subgroups according to the clinical severity (high-risk or non-high-risk) of AMI and subtype of AMI (STEMI or NSTEMI).

Subgroups	Albumin < 3.8 g/dL			3.8 ≤ Albumin < 4.2 g/dL			Albumin ≥ 4.2 g/dL (Reference)
	HR	95% CI	*p*	HR	95% CI	*p*	HR
High-risk group (*n* = 611)							
Crude	1.87	1.17–3.02	0.001	1.59	0.95–2.66	0.066	1.00
Model 1	2.38	1.38–4.36	0.001	1.17	0.65–2.18	0.592	1.00
Model 2	2.38	1.35–4.46	0.002	1.11	0.59–2.16	0.740	1.00
Model 3	3.96	1.52–11.4	0.004	1.15	0.45–3.14	0.763	1.00
Model 4	6.41	2.01–21.1	<0.001	2.30	0.78–7.36	0.128	1.00
Non-high-risk group (*n* = 1642)							
Crude	6.10	4.07–9.15	<0.001	2.79	1.79–4.36	<0.001	1.00
Model 1	4.74	2.98–7.70	<0.001	1.84	1.15–2.99	0.011	1.00
Model 2	4.40	2.67–7.40	<0.001	1.45	0.86–2.45	0.156	1.00
Model 3	2.22	0.75–6.32	0.146	1.72	0.76–3.99	0.191	1.00
Model 4	2.54	1.02–6.61	0.044	2.11	0.68–6.54	0.193	1.00
STEMI (*n* = 1558)							1.00
Crude	5.45	3.82–7.93	<0.001	2.33	1.58–3.47	<0.001	1.00
Model 1	3.67	2.46–5.57	<0.001	1.60	1.07–2.43	0.002	1.00
Model 2	3.40	2.22–5.28	<0.001	1.49	0.97–2.30	0.069	1.00
Model 3	3.47	1.71–7.14	<0.001	1.89	1.03–3.53	0.038	1.00
Model 4	3.75	1.70–8.68	0.001	3.51	1.31–9.64	0.012	1.00
NSTEMI (*n* = 695)							1.00
Crude	14.8	6.80–39.1	<0.001	6.61	2.82–18.1	<0.001	1.00
Model 1	11.0	4.79–29.8	<0.001	4.09	1.66–11.6	<0.001	1.00
Model 2	11.0	4.72–30.3	<0.001	2.74	1.05–8.12	0.038	1.00
Model 3	12.0	3.38–57.8	<0.001	3.02	0.68–17.5	0.149	1.00
Model 4	7.04	1.21–63.8	0.0284	6.45	0.01–2201	0.531	1.00

The high-risk group was defined as a group in which patients had at least one risk factor (acute myocardial infarction due to left main trunk, brain natriuretic peptide >200 pg/mL, peak creatine phosphokinase >8000 IU/L, estimated glomerular filtration rate <30 mL/min/1.73 m^2^, and left ventricular ejection fraction <35%) on admission, while the non-high-risk group was defined as a group in which patients did not have any of those risk factors on admission. CI, confidence interval; HR, hazard risk. Model 1; adjusted for age and sex. Model 2; adjusted for Model 1 plus body mass index and coronary risk factors (hypertension, dyslipidemia, diabetes mellitus, smoking, family history of cardiovascular disease). Model 3; adjusted for Model 2 plus onset-to-admission time, pre-TIMI grade, percutaneous coronary intervention, max creatine kinase, length of hospital stay and statin use at discharge. Model 4; adjusted for Model 3 plus other medication use at discharge (antiplatelet, β-blocker, angiotensin-converting enzyme inhibitor, angiotensin II receptor blocker, mineralocorticoid receptor antagonist and diuretic), left ventricular ejection fraction at acute phase, cardiogenic shock, high-sensitivity troponin T, C reactive protein, alanine aminotransferase as liver function and Killip ≥III. AMI—acute myocardial infarction.

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
