# Peer review of "Prognostic Impact of Serum Albumin for Developing Heart Failure Remotely after Acute Myocardial Infarction"

_nutrients, 2020, doi:10.3390/nu12092637_

Round 1
Reviewer 1 Report
This interesting paper addresses the prognostic value of Serum Albumin after Acute Myocardial infarction.
The method is accurate and the manuscript well written.
However, some revisions could be suggested:
Major revision:
- As relevantly written in the introduction, low serum albumin is both an indicator of poor nutritional status but also of other pathologies. Besides malnutrition, two causes of hypoalbuminemia are frequently encountered in clinical practice: inflammation and hemodilution (edema). Both are also associated with poor prognosis.
To assess whether low albumin is independently associated with worse outcome, I suggest the authors include C Reactive protein or another inflammatory biomarker in the multivariate model, as well as congestive status, if available.
- Moreover, to better control for cardiovascular confounders, I would consider adjustment on GRACE prognostic score (or at least KILLIP class).
- Is the prognostic value of albumin also found after stratification on STEMI or NSTEMI?
Minor revisions :
- I found the title redundant. I suggest the authors suppress the second part.
- Figure 3: Title is not clear to me: AUC for predicting what ? Moreover, the curve announced in the text (“AMI due to left main trunk”) is not shown and should be suppressed in the text.
I suggest the authors add the ROC curve of albumin considered as a continuous variable.
Please add the 95% confidence intervals of the AUC.
- In the Universal definition of MI, troponin should be preferred to CPK. How many patients have had a troponin dosage? I suggest the authors consider troponin and not CPK in the multivariable model and in the figure 3.
Reviewer 2 Report
Thank you for the opportunity to review this paper.
The authors present a paper with focus on the impact of low serum albumin on prognosis after myocardial infarction. One obvious strength is the study size and consecutive patients which in relation to the study topic is unusual. Increasing scientific focus on the frail geriatric patient makes this study a welcome contribution to the scientific context.
Major concerns:
The authors draw far-reaching conclusions from the data concluding that serum albumin is an independent risk factor. However, from reading the paper I am not fully convinced. It is clear from data in table 1 that the charcteristics is very different in the groups. Furthermore there is a clinical interaction between albumin and many of these variables. All this makes statistical analysis very challenging. When reading the methods I still don’t fully understand the modelling of the multivariate analysis presented in Table 2 despite the clearification in section 3.4. From my point of view the multivariate analysis is not taking into account most of the characteristics for the different groups (just look at table 1 medication at discharge, malignancy etc). Why not just do an univariate analysis and then proceed to multivariate?
This must be clearified and more understandable to the reader. A statistical review in necessary.
Minos comments:
Liver function is associated with serum albumin. Why is that not included in the study?
How are the cut off values for serum albumin set?
Why is not medications at discharge in the multivariate model?
The conclusion is too definite in regard to study findings, se above.
Round 2
Reviewer 1 Report
The authors adequately answer to previous comments and significantly improve the manuscript.
Author Response
Thank you very much for your encouraging comment on this study.
Reviewer 2 Report
Thank you for the comments replies. I have no further comments.
Author Response

(The authors gave the same response as above.)
